# Role of the Intestinal Microbiota in the Genesis of Major Depression and the Response to Antidepressant Drug Therapy: A Narrative Review

**DOI:** 10.3390/biomedicines11020550

**Published:** 2023-02-14

**Authors:** Tiziana Mundula, Simone Baldi, Elisabetta Gerace, Amedeo Amedei

**Affiliations:** 1Department of Experimental and Clinical Medicine, University of Florence, 50134 Florence, Italy; 2Department of Health Sciences, Clinical Pharmacology and Oncology Unit, University of Florence, 50139 Florence, Italy; 3Interdisciplinary Internal Medicine Unit, Careggi University Hospital, 50134 Florence, Italy

**Keywords:** pharmacomicrobiomics, depression, antidepressant therapy, gut microbiota, personalized medicine

## Abstract

A major depressive disorder is a serious mental illness characterized by a pervasive low mood that negatively concerns personal life, work life, or education, affecting millions of people worldwide. To date, due to the complexity of the disease, the most common and effective treatments consist of a multi-therapy approach, including psychological, social, and pharmacological support with antidepressant drugs. In general, antidepressants are effective in correcting chemical imbalances of neurotransmitters in the brain, but recent evidence has underlined the pivotal role of gut microbiota (GM) also in the regulation of their pharmacokinetics/pharmacodynamics, through indirect or direct mechanisms. The study of these complex interactions between GM and drugs is currently under the spotlight, and it has been recently named “pharmacomicrobiomics”. Hence, the purpose of this review is to summarize the contribution of GM and its metabolites in depression, as well as their role in the metabolism and activity of antidepressant drugs, in order to pave the way for the personalized administration of antidepressant therapies.

## 1. Introduction

Major depressive disorder (MDD), commonly known as depression, is a serious mental illness characterized by a pervasive low mood that persists for at least two weeks. MDD determines the loss of interest or pleasure in normally enjoyable activities, which can negatively affect personal life, work life, or education, leading to social impairment followed by general health problems and the risk of suicide [1]. The World Health Organization (WHO, Geneva, Switzerland) has estimated that about 350 million people are affected by MDD and considering its incidence of 6%, it is considered the second leading cause of disability worldwide, with high economic and social costs [2]. In addition, this scenario has been exacerbated by the COVID-19 pandemic, with an increased burden of 25% mainly related to isolation and fear of infection [3].

Generally, depression is believed to be caused by a combination of genetic, environmental, and psychological factors, including family history, major life changes, certain treatments, chronic health problems, and substance use disorders [4]. However, the etiology of MDD is still unknown.

Due to the complexity of the disease, the current most common treatment consists of a multi-therapy approach, including psychological, social, and pharmacological support to relieve the manifestations and restore the health state. Antidepressant drugs, in particular, are recommended as an initial treatment choice in people with mild, moderate, and severe depression [1]. Antidepressants, which act by correcting chemical imbalances of neurotransmitters in the brain, are usually very effective; however, their use requires caution [5]. In fact, antidepressant drugs must be taken daily for 1 or 2 weeks to exert their beneficial clinical effects [6], and about 10–30% of the patients show a partial response or do not respond to treatment [7], losing the therapeutic effects with long-term administration (tachyphylaxis) [8]. Lastly, toxicity, side effects, and interactions with other drugs can also limit patients’ compliance [1]. Considering that, to date, antidepressants are among the most prescribed drugs, the establishment of targeted and personalized antidepressant treatments is necessary to improve their safety and efficacy [9,10]. In general, the advancement of pharmacokinetics/pharmacodynamics studies of antidepressant drugs can improve both the quality of life of patients and the sustainability of healthcare systems [11].

In the last few years, scientific evidence has been reported suggesting that a normal composition of the gut microbiota (GM) is required to preserve host health, whereas its alteration may have a role in determining local and systemic pathological conditions, including mood disorders [12]. In addition, GM may represent the first contact point between oral medications and the host body and may regulate both pharmacokinetics/pharmacodynamics pathways through indirect or direct mechanisms; the study of these complex interactions between GM and drugs is currently under the spotlight, and it has been recently named “pharmacomicrobiomics” [12,13].

In this scenario, the purpose of this review is to analyze the available scientific literature and define the contribution of GM and its metabolites in depression development, as well as its role in the metabolism and activity of antidepressant drugs, to pave the way to personalized administration of antidepressant treatments.

## 2. Literature Search

A computerized search of the articles published until November 2022 was conducted in PubMed and Google Scholar, using the following search string: (gut microbiota) AND (selective serotonin reuptake inhibitors OR serotonin-norepinephrine reuptake inhibitors OR tricyclic antidepressants OR monoamine oxidase inhibitors OR atypical antidepressants). Additional papers were identified by reviewing the reference lists of relevant publications. Publications with relatively low reliability and written in another language than English were excluded. The data were extracted based on their relevance to the topic.

## 3. Types, Symptoms, and Causes of Depression

MDD is a heterogeneous and complex disease that can have devastating effects on an individual’s function and life quality. It is characterized by several symptoms, including a sad mood, pessimistic worries, anhedonia, sleep alterations, poor focus, changes in appetite, fatigue, psychomotor agitation or retardation, and the risk of suicide. To be diagnosed with depression, the aforementioned symptoms must be present for at least two weeks. Different types of depression are usually classified as major depression, persistent depressive disorder, perinatal depression, seasonal affective disorder, and depression with symptoms of psychosis. All these forms of depression may be categorized as mild, moderate, or severe based on the frequency and intensity of the symptoms. The disorder’s course widely varies from one episode lasting months to a lifelong disorder with recurrent major depressive episodes [1].

Even though the depression mechanisms have not been completely elucidated yet, dysfunctions in serotoninergic (5-hydroxytryptamine, 5-HT), dopaminergic (DA), noradrenergic (NE), and gamma-aminobutyric acid (GABAergic) neurotransmissions have been considered a plausible explanation of the pathophysiology of depression over several decades [14,15]. Furthermore, several factors, such as the hypothalamic-pituitary axis, host genetic polymorphisms, environmental factors, but also neurological, hormonal, immunological, and neuroendocrinological mechanisms, seem to play a role in the development of the depression phenotype. Psychological stress and adversity associated with genetic variants of monoaminergic transporter genes or defined receptor genes also appear to play a specific role in vulnerability to depression [16]. Some studies suggest that a predisposition to the disease occurs early in infancy and even in utero (perinatal depression). Indeed, the deprivation of maternal care can reflect itself in the epigenetic alteration of glucocorticoid receptors in the hippocampus, increasing the activation of the hypothalamic–pituitary–adrenal axis in response to stress factors and predisposing to depression later in life [17]. Another attractive theory on the development of depression is the “cytokines model,” which supposes that immunity plays a role in the modulation of brain structure and function through the cytokines’ production. Indeed, depressed patients often show a high circulating level of proinflammatory cytokines such as interleukin (IL)-1, IL-2, IL-6, tumor necrosis factor alpha (TNF-α) and C-reactive protein (CRP), supporting the hypothesis that a chronic production of proinflammatory cytokines may provoke neurotransmitter imbalance leading to neuropsychiatric diseases [18,19].

In this scenario, considering the documented capability of the GM to modulate the immune system and regulate brain functions through the so-called “gut-brain axis” [20], it has been proposed that the GM has a role in the pathogenesis of depression [21]. Indeed, it has been demonstrated that fecal microbiota transplantation (FMT) from depressed patients to microbiota-depleted rats can induce behavioral and physiological features characteristic of depression in the recipient animals [19]. In addition, several studies have documented a profound GM dysbiosis in depressed patients [22,23] and some authors have suggested that a predisposition to depression occurs early in infancy and even in utero. Indeed, during pregnancy, stress-induced changes to both maternal vaginal and intestinal microbiota are associated with changes in reproductive hormones, stress hormones, and neurosteroids that can be transferred to offspring in utero or during parturition, determining a long-term risk for neurobehavioral disorders [24].

## 4. The Role of the Gut-Brain Axis in Depression

The GM is composed of more than 100 trillion microorganisms carrying three times the number of human genes and, under normal conditions, is involved in several physiological processes such as food digestion, vitamin synthesis, the regulation of intestinal barrier homeostasis, and immunity modulation [25].

Conversely, harmful changes in the GM function and composition determine an intestinal dysbiosis that may contribute to the genesis of local or systemic diseases and eventually result in a possibly permanent alteration in the physiological response in a way that is mainly dependent on inflammatory processes [26,27]. Interestingly, a recent body of evidence documented the existence of the “gut-brain axis,” a complex bidirectional system in which communication occurs through three parallel and interplaying pathways that involve nervous, endocrine, and immune signals [28]. Different preclinical and observational studies have documented that GM has a prominent role in mediating brain functions via the gut-brain axis [20], and its importance in the pathogenesis of brain disorders, including depression, has been proposed [21,29].

The main mechanism responsible for the effect exerted by alterations in GM on distant body districts, including the brain, is the modification of intestinal permeability. In fact, in GM dysbiosis, pathogens’ overgrowth promotes the loss of the intestinal barrier, determining a “leaky gut” condition that allows intestinal microorganisms, and especially their metabolites, to enter the systemic circulation by crossing the intestinal barrier [30].

Thus, regarding the gut-brain crosstalk, microbial-derived metabolites are also involved in the afferent input of the vagus nerve [31], in the stimulation of the enteric nervous system and in the regulation of the hypothalamic-pituitary-adrenal axis [32].

Overall, the main metabolic products involved in the communication between the gut and brain and whose alterations may have a role in the genesis of MDD are neurotransmitters, short-chain fatty acids (SCFAs), lipopolysaccharides (LPS), and formyl peptides. 

### 4.1. Neurotransmitters

Neurotransmitters play a fundamental role in “gut-brain axis” signaling, and intestinal bacteria may react to the host’s neurochemical compounds or produce their own. It was largely demonstrated that GM can produce 5-HT, NE, DA, melatonin, histamine, tyramine, phenylethylamine, glutamate, and GABA neurotransmitters [33,34,35]. In vitro studies have documented the existence of a great number of microbes producing neurotransmitters [35]. On the other hand, experimental evidence showed that germ-free mice exhibited a severe imbalance in cerebral neurotransmitters, with high noradrenaline and reduced GABA and serotonin levels [36]. In addition, preclinical and clinical studies have also confirmed that the GM manipulation of probiotics, especially *Lactobacillus* and *Bifidobacterium* strains, can improve depressive symptoms by increasing 5-HT levels in the brain [37,38].

An essential element in neurotransmitter synthesis is the availability of precursors, usually represented by amino acids (AAs), whose levels are strictly related to the diet [39]. During their path, particularly in the colon, the AAs undergo various stages and are directly metabolized by gut bacteria [40,41]. For example, the fermentation of basic AAs leads mainly to decarboxylate metabolites, while the catabolism of arginine can lead to agmatine, putrescine, spermidine, and spermine through the polyamine pathways [33]. Arginine can also enter the arginase pathway of some bacteria, where it is converted into urea and ornithine and subsequently catalyzed into glutamate [42]. Afterwards, the glutamate can be deaminated by the intracellular enzyme glutamate decarboxylase into GABA [33,42]. Instead, histidine metabolism can produce histamine, a neurotransmitter implicated in allergic reactions but also neurological and psychiatric diseases, while the catabolism of tyrosine determines the production of tyramine, phenols, and p-coumarate. Importantly, tyrosine is a precursor of catecholamines, and tyramine is a neurotransmitter involved in the side effects of MAOI antidepressants [33]. In addition, phenylalanine catabolism can produce phenylethylamine [43] (PEA) and trans-cinnamic acid. In particular, PEA has been found in the brain in low concentration and seems to stimulate the release of 5-HT by acting on its transporter [44].

Anyway, the most important AA for gut-brain communication is tryptophan, an essential aromatic AA that can be transformed into several metabolites such as serotonin, kynurenine, or indole. Approximately 90% of tryptophan is degraded in the liver through the “kynurenine pathway,” but several gut bacteria can modulate this process, mainly through gut metabolites, for instance SCFAs, resulting in higher tryptophan levels [45]. About 5% of tryptophan can be converted directly by GM into indole compounds through the indole pathway [46]. In detail, the indole is produced by specific bacteria, such as *Echerichia coli*, through a transformation dependent on tryptophan-catabolizing enzymes [47]. In the liver, the indole is subsequently converted into several metabolites, some of which, e.g., indoxyl sulphate, show anxiogenic effects, while others, such as isatin, have an anxiolytic effect. Indeed, isatin activates the vagus nerve and travels to the brain, where it acts as a potent MAO-B inhibitor, increasing DA levels [47,48]. Finally, a tryptophan fraction is converted into 5-HT and melatonin through the serotonin pathway [49].

In conclusion, the control of the neurotransmitters’ production by GM involves an intricate network of pathways, mostly under exploration, that will require deeper investigation in the future.

### 4.2. Short Chain Fatty Acids

The SCFAs are monocarboxylic acids predominantly produced in the colon through the fermentation of dietary polysaccharides by the GM, mainly from anaerobic bacteria [50,51,52]. Locally, they represent an energy source for colonocytes and regulate intestinal barrier integrity; however, unmetabolized SCFAs can cross the intestinal barrier and enter the systemic circulation, acting as signals for host metabolic, immune, and neurocognitive functions [53]. In detail, growing evidence suggests that an alteration of the intestinal SCFA abundance is associated with the reduction of several neurotransmitters in the brain, determining a depressive phenotype [54,55]. In general, the SCFAs affect brain functions through several direct and indirect pathways. Firstly, they directly contribute to maintaining the integrity of the brain-blood barrier (BBB) and enhance the Claudin5 expression, both of which are essential to protect the brain from inflammatory cytokines and toxins derived from systemic circulation [51,56]. Recent findings have documented that a loss of BBB integrity is evident in germ-free mice and mood disorders, suggesting that normal BBB tightness is fundamental to protecting against depression [56]. Moreover, after crossing the BBB, the SCFAs can stimulate the microglia, regulating their functions [57]. In addition, SCFAs can display indirect activity on the brain through the gut. For instance, the SCFAs bind to free fatty acid receptor 2 (FFA2) expressed on the colonic enteroendocrine L-cells, stimulating the release of gut hormones such as glucagon-like peptide 1 (GLP-1), which may interact with GLP-1 receptors, which have been recently identified in neurons of the amygdala, hippocampus, and dorsal raphe nucleus [58]. Importantly, the activation of these receptors can exert an antidepressant effect through several mechanisms, including inhibition of neuroinflammation, promotion of neurogenesis, and neurotransmitter production [59].

### 4.3. Lipopolysaccharides

Lipopolysaccharides (LPS) are structural components of Gram-negative bacteria walls that exert pro-inflammatory effects by activating the Toll-like receptor (TLR)-4 and by triggering NF-kB (nuclear factor kappaB) [60]. The gut is the main LPS source in the human body, and, while in physiological conditions the LPS load is well tolerated, its high content, determined by the disruption of gut barrier integrity, leads to “metabolic endotoxiemia”, which is associated with a chronic systemic low-grade inflammation and several related pathologies [61]. In detail, several reports have demonstrated the association between high LPS levels and depression, especially for its ability to stimulate the production of proinflammatory cytokines and to activate indole and GluN2B receptors [62,63].

### 4.4. Formyl Peptides

The formyl-peptide receptors (FPRs) are transmembrane G protein-coupled receptors that can be considered pattern recognition receptors, interacting with chemotactic factors released by bacteria or damaged host tissues. The FPRs are located on the surface of immune cells and are correlated with host innate defense mechanisms [64]. To date, three types of FPR have been identified (FPR1, FPR2 and FPR3), which can have pro-inflammatory or anti-inflammatory effects depending on ligands and the environment [65]. Particularly, the microbial-derived formyl peptides can activate FPRs, which are expressed by both the enteric nervous system and the central nervous system. Interestingly, recent evidence showed that formyl peptides regulate neuroinflammation and emotional behaviors, preventing neurodegenerative diseases and mental disorders such as anxiety and depression [65,66,67].

## 5. Antidepressants Therapeutic Approaches

The therapy of depression is complex and includes multi-therapy approaches such as psychological, social, and pharmacological treatments. In cases of drug resistance, options may include a combination of electroconvulsive therapy (ECT), transcranial magnetic stimulation (TMS), or light therapy. Antidepressant drugs are recommended as an initial treatment option for people suffering from mild, moderate, and severe depression [68].

The antidepressant drugs can be divided into five main classes according to their different mechanisms of action: selective serotonin reuptake inhibitors (SSRIs), serotonin-norepinephrine reuptake inhibitors (SNRIs), tricyclic antidepressants (TCAs), monoamine oxidase inhibitors (MAOIs), and atypical antidepressants (Table 1) [69].

In general, antidepressants alleviate depressive symptoms by acting on neurotransmission, particularly in serotoninergic, noradrenergic, and GABAergic systems, but they can also influence neurogenesis-related processes in specific brain areas [70]. The clinical effects of antidepressant medications appear after a minimum of 3–4 weeks of treatment, and disease control requires long-term maintenance administration for up to 6–12 months [71]. Even though antidepressants are well absorbed after oral intake and their availability is reduced by their metabolism in the liver, each class has different pharmacokinetic variables, interactions with other drugs, and side effects [72]. Moreover, antidepressants can be used in monotherapy or in combination to enhance efficacy or reduce negative effects [73]. Although antidepressants are usually very effective, their use requires some caution. In fact, over a third of patients with MDD show drug resistance, and some therapies have heavy side effects [74].

Alterations in GM may also play a role in causing resistance to antidepressant drugs. In fact, Belzeaux and colleagues reported that the microbial compositional and functional signatures of MDD patients, either at baseline or after antidepressant treatment, differed significantly between patients responding or non-responding to antidepressant drugs, suggesting that alterations in GM composition and metabolic function are relevant in determining the response to antidepressants [75] (Figure 1).

Therefore, the efforts of scientists are currently focused on improving the effectiveness and tolerability of drugs as well as the promptness of symptom relief [74,76], considering that the selection of pharmacological therapy needs to be personalized based on the peculiar characteristics of the drug and patient [77].

### 5.1. SSRIs

To date, SSRIs are the most commonly prescribed antidepressant drugs and, due to their safety profile, are considered the first-choice treatment for moderate and severe depression [77]. The SSRIs approved by the Food and Drug Administration (FDA, Silver Spring, MD, USA) to treat depression are paroxetine, fluoxetine, sertraline, fluvoxamine, citalopram, and escitalopram, which have different chemical structures but a similar profile of action [78]. Their mechanism is mainly based on blocking the 5-HT reuptake in the synaptic 5-idrossitriptamine transporter (SERT) and consequentially, enhancing the serotoninergic transmission. SSRIs are metabolized in the liver by cytochrome P-450, and their half-lives are between 21 h (for paroxetine) and 4 days (for fluoxetine) [78]. The SSRIs’ treatment starts at a low dose, and, after some weeks, the dosage must be adjusted and incremented gradually. The main reported adverse effects include nausea, vomiting or diarrhea, headache, drowsiness, insomnia, sedation, cognitive impairment and sexual problems, but they also impact appetite, leading to weight loss or weight gain [79]. The SSRIs’ polytherapy needs caution because it may cause the “serotonin syndrome,” which is determined by high levels of 5-HT accumulation, and it is characterized by symptoms such as hyperthermia, hypertension, confusion, tremors, coma, and even death [80]. In addition, patients below 25 years must be monitored because of the high reported risk of increased suicidal thoughts during treatment and of “discontinuation syndrome,” which occurs when the therapy is abruptly stopped [81].

### 5.2. SNRIs

The SNRIs are monoamine reuptake inhibitors, specifically inhibiting the reuptake of 5-HT and NE, and are effective in patients who have had unsuccessful treatment with SSRIs. There are eight FDA-approved SNRIs in the United States, with venlafaxine being the first drug developed in 1993 and levomilnacipran being the latest drug to be developed in 2013, but this family also includes desvenlafaxine, duloxetine, sibutramine, tramadol, venlafaxine, and milnacipran [69,82]. In comparison with SSRIs, this class of antidepressant drugs shows a minor half-life, varying from 8 to 14 hours, and few or no active metabolites. In fact, only venlafaxine has an active metabolite called “Desvenlafaxine,” which is believed to work similarly, though some evidence suggests lower response rates compared to venlafaxine [83]. The side effects of SNRIs are similar to those reported for SSRIs, including “serotonin syndrome” and the risk of suicide behaviors in young people.

### 5.3. Tricyclic Antidepressants (TCAs)

The TCAs are among the earliest antidepressants to be developed. The cyclic antidepressants are designated as tricyclic or tetracyclic, depending on the number of rings in their chemical structure, and include amitriptyline, clomipramine, imipramine, trimipramine, nortriptyline, maprotiline, protriptyline, desipramine, amoxapine, and doxepin. TCAs compounds not only modulate serotonergic and noradrenergic systems (5-HT and NE reuptake inhibition), but also modify antihistaminic, alpha cholinergic, and muscarinic receptors [84]. Due to their poor selectivity, the TCAs’ use is limited because of their toxicity, contraindications, risk of intentional overdose, and drug interactions. Their side effects depend on the type and strength of receptor affinities and include symptoms such as orthostatic hypertension, weight gain, constipation, sexual dysfunction, seizures, blurred vision, tachycardia, and the risk of suicide.

The TCAs are commonly well absorbed after oral ingestion, metabolized in the liver by cytochrome P-450, and then excreted in the urine [81]. However, about 7% of patients are slow metabolizers of TCA (the CYP2D6 isoenzyme), and so they are exposed to more severe adverse effects. In addition, the concomitant use of TCAs with MAOIs, SSRIs, and NSRIs is not recommended [69]. Due to their unfavorable profile, they are not indicated as first-line treatment for depression, and they cannot be used in patients with a history of cardiac disease because TCAs can induce QT interval prolongation, ventricular fibrillation, and sudden cardiac death. For these reasons, TCAs have been replaced by a new class of antidepressants; however, they are effective in some patients at relieving depression when other treatments have failed [84].

### 5.4. Monoamine Oxidase Inhibitors (MAOIs)

The MAOIs were the first class of developed antidepressants, and they act by inhibiting the activity of both monoamine oxidase enzymes (MAO-A and MAO-B), thus preventing the breakdown of monoamine neurotransmitters (5-H7, histamine, DA, NE, and epinephrine) and thereby increasing their availability. In particular, MAO-A preferentially deaminates 5-HT, melatonin, epinephrine, and NE, while MAO-B preferentially deaminates phenethylamine. MAOIs are classified into nonselective MAO-A/MAO-B inhibitors (iproniazid, isocarboxazid, hydracarbazine, phenelzine, tranylcypromine), selective MAO-A inhibitors (bifemelane, methylthioninium chloride, moclobemide, pirlindole), and selective MAO-B inhibitors (rasagiline, selegiline, safinamide) [85]. Like other antidepressant drugs, they have a latency time response, generally established at 2 weeks. Understandably, this class of antidepressant drugs is not the first line of treatment for depression because of their low safety profile and their remarkable side effects (including sexual dysfunction, insomnia, headaches, and weight gain, but also food interactions). For example, the “cheese reaction” is the most commonly reported adverse food reaction that occurs when the MAOIs are taken together with foods rich in tyramine, like aged cheeses, smoked or pickled meats, or fish, provoking a hypertensive crisis [86]. The MAOIs can also be responsible for the “serotonin syndrome” when co-administered with SSRIs and a “discontinuation syndrome” if suddenly stopped. However, the MAOIs appear to be particularly effective in the management of bipolar depression, according to a retrospective analysis from 2009 and are also effective for treatment-resistant depression and atypical depression [19]. Importantly, the MAOIs can be used to treat Parkinson’s disease by selectively targeting MAO-B, as well as providing an alternative for migraine prophylaxis. In fact, newer MAOIs such as selegiline (typically used in the treatment of Parkinson’s disease) and the reversible MAOI moclobemide provide a safer alternative [20] and are now sometimes used as first-line therapy.

### 5.5. Atypical Antidepressants

Atypical antidepressants are new drugs that are not included in the four classes described so far since they have different mechanisms of action. To date, the atypical antidepressants approved by the FDA are bupropion, mirtazapine, nefazodone, trazodone, vilazodone, and vortioxetine. Bupropion is a DA and NE reuptake inhibitor and blocks several nicotinic receptors. As with other antidepressants, bupropion may induce several side effects like insomnia, agitation, headache, dry mouth, nausea, and seizures [87]. Mirtazapine, on the other hand, has a double action. It acts on both serotonergic and noradrenergic systems, but it is also a potent histamine (H1) receptor antagonist. It is metabolized mainly in the liver and excreted by the kidneys, and the main characteristic of this drug is its rapid action onset (less than 1 week) [88]. Instead, nefazodone enhances serotonergic transmission and antagonizes the α(1)-adrenergic receptors; it has a low bioavailability (20%) and a high liver toxicity. Trazodone is a multimodal antidepressant drug that, in addition to its inhibitory activity on SERT, is a competitive ligand at 5-HT1A, 5-HT2A, 5-HT2C receptors, and α1-adrenoceptors [89,90]. To date, the full spectrum of trazodone’s actions has not been completely established, and it has been hypothesized that trazodone may have multiple pharmacological effects, including modifying glutamatergic excitatory neurotransmission [91]. A peculiar characteristic of this drug is that trazodone does not display the typical side effects of SSRIs and SNRIs. However, it can prolong the cardiac QT interval and may induce priapism and drowsiness [92]. Vilazodone is a selective partial agonist and reuptake inhibitor (SPARI) with minor adverse effects in comparison to SSRIs; it has a half-life of 25 hours and a high bioavailability (72%). It is metabolized in the liver mainly by cytochrome P-450 (3A4 isoenzyme). Like other SSRIs, it may be responsible for “Serotonin Syndrome” and adverse effects such as nausea, insomnia, and decreased libido [93]. Generally, atypical antidepressants are frequently administered to patients with major depression who have inadequate responses or intolerable side effects during first-line treatment with SSRIs. However, atypical antidepressants are often first-line treatments if the drug has a desirable characteristic (e.g., sexual side effects and weight gain occur less often with bupropion than with SSRIs).

## 6. Gut-Microbiota Based Pharmacokinetics

Usually, orally administered drugs experience an intensive first-pass metabolism by the GM, and this activity has a profound effect on their pharmacokinetic profiles [94]. A recent study has demonstrated that the availability of some drugs can be reduced in the presence of specific gut bacteria through a phenomenon called “bioaccumulation”, e.g., the antidepressant duloxetine can be englobed by certain bacteria, reducing its pharmacological effects [95,96]. In addition, GM influences the drug’s absorption into the bloodstream. Drug transporters localized in the enterocytes‘ walls are transmembrane proteins that regulate the influx/efflux of chemicals ingested through the oral route [62,95,97] and their activity is influenced by many environmental factors such as hypoxia, genetic polymorphisms, and concomitant pathologies [98,99]. However, GM can act on a drug with an epigenetic control or a direct interaction [100]. For instance, the multidrug transporter ABCB1/MDR1 P-glycoprotein, which binds substrates belonging to several pharmaceutical classes (including antidepressants) and plays a pivotal role in drug toxicity due to its ability to efflux these molecules back into the lumen, is strongly regulated by histone deacetylase inhibitors as microbial-derived SCFAs [101,102].

The epigenetic control by gut metabolites can also act at distal sites, on membrane transporters of liver or renal cells, modifying the absorption or elimination of several toxins and drugs; thus, liver detoxification activity and excretion processes in kidneys are sensitive to microbial-derived molecules [103].

Another fascinating mechanism by which GM can modulate drug transport is the direct interaction of gut metabolites with drug transporters. Recent evidence has demonstrated that butyrate (an important SCFA) availability is essential to limit the potential hepatic toxicity of antidepressants because it regulates the hepatic organic anion transporter OAT7, which operates by exchanging butyrate entering hepatocytes with sulphate-conjugate drug metabolites released in blood plasma [100].

Intriguingly, the GM-derived metabolites can also affect drug influx into or efflux from the brain parenchyma. Many antidepressants, being small lipophilic molecules (amitriptyline, fluoxetine, paroxetine, citalopram), easily cross the BBB by passive diffusion, but, at the same time, they are substrates of efflux pumps expressed by the BBB [104].

Therefore, since microbial metabolites such as SCFAs and LPS modulate the BBB permeability, it is likely that the antidepressants’ availability also depends on the BBB control operated by the GM [105,106].

Another mechanism by which GM may affect drug pharmacokinetics is the direct metabolism of microbial enzymes. Although pharmacology dogma holds that the first step of drug metabolism mainly happens in the liver, current evidence has documented that it occurs instead in the intestine, with high intra- and inter-individual variability. The GM is a remarkable source of enzymes that can transform the drugs into active or inactive metabolites; for example, fluoxetine (an SSRI antidepressant) can be biodegraded (in a range from 48% to 85%) by a specific consortium of bacteria [107]. Regarding antidepressant therapies, Yu and colleagues have demonstrated that the degradation exerted by the GM is the main reason for the low bioavailability of paeoniflorin, a chemical compound that showed remarkable antidepressant effects [108]. Moreover, Lukić et al. have shown that the dysregulation of a single bacterium, *Ruminococcus flavefaciens*, can dramatically reduce the clinical response to duloxetine in depression [109].

## 7. Modulation of GM by Antidepressants

Several drugs have been proven to have a marked effect on GM, and antidepressants modify its composition, function, and diversity by exerting antibacterial (bacteriostatic or bactericidal in a dose-dependent manner) or antifungal properties [110]. For instance, different studies have reported that depressed patients displayed a dysbiotic gut mycobiome with increased levels of *Candida albicans* [111,112], an opportunistic fungal pathogen able to produce some toxic substances for the host, such as acetaldehyde [113]. Since acetaldehyde is transformed by the aldehyde dehydrogenase 2 into acetate, a metabolite that can easily cross the BBB, altering the brain’s neurotransmission [114], it is tempting to speculate that the antidepressant mechanisms of some drugs (e.g., sertraline, fluoxetine, doxepin, imipramine, and nortriptyline) could be related to their antifungal activity against *C. albicans* [115,116]. Moreover, in addition to their in vitro antimicrobial effects, in vivo studies have also confirmed that antidepressants act on some specific genera or species, changing the richness of the GM [117]. For instance, citalopram and phenelzine have been shown to be active on the strain *Faecalibacterium prausnitzii*, while desipramine influences *Akkermansia muciniphila* levels [115]. In addition, paroxetine has reported a strong bactericidal action on various Gram-positive and Gram-negative bacteria, especially *Acinetobacter baumannii* and *Escherichia coli* ATCC 35218 [118].

A recent randomized controlled trial showed that, in depressed patients, escitalopram treatment restored normal GM composition, correcting the disease-related dysbiosis, which is characterized by a higher alfa-diversity [119]. In another study, Zhang et al. showed that fluoxetine and amitriptyline given for 15 weeks to rats previously submitted to mild unpredictable stress modified the GM reduction of the Firmicutes/Bacteroidetes (F/B) ratio, a parameter that has been associated with neurological improvement [110,120].

Furthermore, antidepressants can exert their effects on intestinal bacteria involved in numerous beneficial and anti-inflammatory pathways for the host, and a depletion of these strains may result in chronic systemic low-grade inflammation and weight gain, liver fatty disease, or other metabolic side effects seen during long-term treatment [121,122].

More specifically, the increase in the Porphyromonadaceae family and *Alistipes* spp. (due to prolonged antidepressant administration) was associated with higher gut inflammation and colitis [123,124].

Moreover, Dethloff and colleagues have evaluated the effect of 2 weeks of paroxetine administration on the GM composition of depressed mice, reporting a significant decrease in alfa-diversity and increased levels of bile acids [125]. Primary bile acids are synthesized in the liver from cholesterol, and then, through the bile, they are excreted in conjugated form into the gut, where they are further transformed by the GM into numerous secondary bile acids [126]. Bile acids alter gene expression or activate several nuclear receptors and G protein-coupled receptors in different tissues and organs, and so are involved in the digestion of lipids, the absorption of lipid-soluble vitamins, and drug detoxification [127]. In addition, recent evidence has highlighted the role of bile acids in the modulation of brain signaling pathways by still poorly defined molecular mechanisms, and, therefore, it is tempting to speculate that part of paroxetine’s central effects could be dependent on its effects on bile acid levels [125].

## 8. Emerging Therapies Targeting GM

To date, the huge amount of data obtained thanks to the advent of novel high-throughput DNA sequencing methodologies has clarified not only the crucial role of the GM in the modulation of many hosts’ physiological functions but also its involvement in the pathogenesis of many local and systemic disorders [128]. Therefore, in the last few years, many strategies, such as prebiotic and probiotic supplementation and FMT, have been developed in order to restore intestinal eubiosis and alleviate the symptoms of various gastrointestinal and non-gastrointestinal conditions [129].

For instance, given the well-established presence of bidirectional communication between the gut and the brain, many interventions have been performed in patients with cognitive and neurological disorders, and especially probiotic supplementation and FMT have demonstrated promising results in enhancing patients’ cognitive functions [29].

A systematic review performed by Huang and colleagues has reported that probiotics significantly decreased the depression scale score in MDD patients, confirming their potential use as a preventive strategy for depression [130].

Of note, a novel class of probiotics named “psychobiotics” has been recently introduced, referring to specific formulations that can improve not only gastrointestinal function but also depressive and anxious symptoms by affecting the gut-brain axis through the modulation of immune, humoral, neural, and metabolic pathways [131].

However, although research on the effects of psychobiotics on mental and neurological disorders is increasing, the currently available evidence is still limited, and further studies are needed to better define their mechanism of action and their possible future utilization as novel treatment/prevention tools for various mental and cognitive disorders.

## 9. Conclusions and Future Perspectives

Over the last few years, thanks to the new high-throughput sequencing technologies, the microbiota exploration has rapidly evolved, becoming a hot topic of basic, preclinical and clinical research, highlighting the role of GM as a major factor shaping human physiology. This development also led to the discovery of several two-way communication axes between the intestine and the other organs of the human body, including the brain, which are primarily modulated by biologically active microbial products.

Currently, it is well established that the gut-brain axis is primarily modulated by biologically active microbial molecules and metabolites. Furthermore, the intestinal dysbiosis has been linked to the pathogenesis of different psychiatric and neurological disorders, including the major depressive disorder, by causing detrimental changes in the bidirectional relationship between the GM and the nervous system. Therefore, the development of precision and personalized approaches based on patient microbiota composition could represent a powerful tool to achieve a safe and effective pharmacological treatment of mood disorders, whose prevalence is increasing worldwide. In fact, as discussed in this review, the drugs’ therapeutic response and toxicity can be modulated by GM, which, through its involvement in the processes of drug absorption, distribution, metabolism, and excretion, can remarkably contribute to therapeutic success. Therefore, the progress of the study of drug-microbiota mechanisms of action will be fundamental for the development of new treatments, the modification of the characteristics of some antidepressants already commercially available, and the evaluation of the interactions between different drugs. In addition, considering the antidepressants’ effects on GM composition, the monitoring of the intestinal changes determined by their assumption could allow corrective measures to be taken to avoid gut dysbiosis, drug resistance, iatrogenic pathologies, toxicity, and side effects. In addition, adequate probiotic mixtures might be administered with antidepressants to manipulate the microbiota in order to support the drug treatment, improving the therapeutic outcome.

Hence, a better understanding of the impact of pharmacomicrobiomics in the treatment of depression in the future will provide fundamental information for developing a personalized antidepressant administration.

## Figures and Tables

**Figure 1 biomedicines-11-00550-f001:**
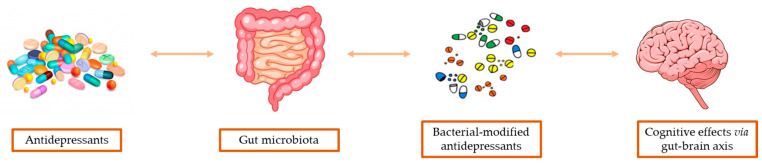
Schematic representation of antidepressant pharmacomicrobiomics.

**Table 1 biomedicines-11-00550-t001:** Classification, mechanisms of action, and side effects of antidepressant drugs.

Classification	Mechanism of Action	Half-Life	Side Effects/Toxicity
SSRIs (Citalopram, Escitalopram, Fluoxetine, Fluvoxamine, Paroxetine, Sertraline)	SERT inhibition, enhancement of serotonergic transmission	1–4 days	Cognitive impairment, nausea, prolonged QT interval, serotonin syndrome, sexual dysfunction, suicidal thoughts, xerostomia
SNRIs (Desvenlafaxine, Duloxetine, Levomilnacipran, Milnacipran, Venlafaxine)	SERT and NET inhibition	8–14 hours	Constipation, high diastolic blood pressure, nausea, serotonin syndrome, sexual dysfunction, suicidal thoughts
TCAs (Amitriptyline, Amoxapine, Clomipramine, Desipramine, Doxepin, Imipramine, Maprotiline, Nortriptyline, Protriptyline, Trimipramine)	Inhibition of SERT, NET, α_1_, α_2_, M_1_ and H_1_ receptors.	1–3 days	Blurred vision, constipation, orthostatic hypertension, seizures, sexual disfunction, suicidal thoughts, tachycardia, weight gain
MAOIs (Isocarboxazid, Phenelzine, Selegiline, Tranylcypromine)	MAO_A_ and MAO_B_ inhibition, increase of 5-HT, histamine, DA, NE and epinephrine levels	2–12 hours	Headache, insomnia, serotonin syndrome, sexual dysfunction, weight gain
Atypical Antidepressants (Bupropion, Mirtazapine, Trazodone, Vilazodone, Vortioxetine)	DAT inhibition, antagonism of α1, 5-HT_2_ and 5-HT_3_ receptors	1–3 days	Abnormal bleeding, agitation, dry mouth, headache, insomnia, nausea, seizures, sexual dysfunction

SSRIs: selective serotonin reuptake inhibitors; SNRIs: serotonin-norepinephrine reuptake inhibitors; TCAs: tricyclic antidepressants; MAOIs: monoamine oxidase inhibitors; SERT: serotonin transporter; NET: norepinephrine transporter; DAT: dopamine transporter; MAO: monoamine oxidase; DA: dopamine; NE: norepinephrine, 5-HT: 5-hydroxytryptamine; M1: muscarinic acetylcholine receptor M1; H1: histamine receptor H1; α1: alpha-1 adrenergic receptor; α2: alpha-2 adrenergic receptors.

## Data Availability

Data sharing not applicable. No new data were created or analyzed in this study. Data sharing is not applicable to this article.

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
