# Peer review of "Role of the Intestinal Microbiota in the Genesis of Major Depression and the Response to Antidepressant Drug Therapy: A Narrative Review"

_biomedicines, 2023, doi:10.3390/biomedicines11020550_

Round 1

Reviewer 1 Report

Dear authors,

After the review process, I have several comments: you should present in the figure legend how it was realized; a correlation between microbiota bioactivity and bioavailability should be added in section 4.5 or into a new section; you should add new comments related to the microbial dysbiosis that can result in a possibly permanent alteration in the physiological response; you should highlight recent contributions made to alleviating human dysbiosis, as future paper valorization in the clinical application.

Best regards!

Author Response

Q/S 1: You should present in the figure legend how it was realized

Q/S 1 Reply: Thank you for your comment, we have prepared the figure by ourselves on PowerPoint, but we think that is not elegant to write it in the figure legend.

Q/S 2: a correlation between microbiota bioactivity and bioavailability should be added in section 4.5 or into a new section

Q/S 2 Reply: Thank you, we have added a comment about the role of the microbiota in the modulation of  drugs’ bioactivity and bioavailability in section 4 (please see lines 278-283)

Q/S 3: you should add new comments related to the microbial dysbiosis that can result in a possibly permanent alteration in the physiological response

Q/S 3 Reply: Thank you, we have added in the text the suggested comment (please see lines 117-120)

Q/S 4:  you should highlight recent contributions made to alleviating human dysbiosis, as future paper valorization in the clinical application.

Q/S 4 Reply: In agreement with the reviewer, we have added a paragraph describing the recent strategies to modulate gut microbial communities in order to restore an intestinal eubiosis condition (please see lines 507-529)

Reviewer 2 Report

The manuscript entitled „ Bugs and drugs: how the intestinal microbiota modulates antidepressant therapies” presents interesting issue, but some problems should be corrected.

Authors prepared a literature review, which intended to answer a specific question – as defined within the title of the study “how the intestinal microbiota modulates anti-depressant therapies”, being a specific problem. However, within their manuscript, Authors presented a lot of information about either intestinal microbiota or depression/antidepressant therapies, but they did not answer this specific question.

It may be observed even based on the sub-chapters, which are organized as follows:

(1)    Types, symptoms, and causes of depression (chapter about depression/antidepressant therapies, but not about intestinal microbiota)

(2)    The role of the gut-brain axis in depression (chapter about intestinal microbiota and depression/antidepressant therapies, but not answering the specific question about how the intestinal microbiota modulates anti-depressant therapies)

(3)    Antidepressants Therapeutic Approaches (chapter about depression/antidepressant therapies, but not about intestinal microbiota)

(4)    Gut microbiota and antidepressant therapies (chapter supposed to answer answering the specific question about how the intestinal microbiota modulates anti-depressant therapies, but as short as 15 lines and referring 6 references and not answering it)

(5)    Gut-microbiota based pharmacokinetics (chapter supposed to answer answering the specific question about how the intestinal microbiota modulates anti-depressant therapies, but as short as less than 1 page and not answering it)

(6)    Modulation of GM by antidepressants (the chapter about the reverse association – instead of how the intestinal microbiota modulates anti-depressant therapies describing how the anti-depressant therapies modulate intestinal microbiota)

The presented manuscript not only does not answer the specific question and the aim is not met, but also there are serious doubts associated with included references and general content.

The serious flaw of the presented manuscript is associated with the fact, that it presents a highly subjective review, not a systematic review. While the systematic review has a key role for broadening knowledge, the other reviews don’t have such role.

Taking into account, that the Materials and methods section is not presented (it should be added), without any specific information, it is hard to understand which studies were included into review and why. Authors did not present any key words, which were used during literature search, inclusion and exclusion criteria of references, information about the procedure of literature search conducted by them, number of chosen references, as well as information if some of them were excluded from the review and on the basis of which criteria. As a number of recent publications that are related to the issue were not included, it is a serious problem.

Authors do not present the current and comprehensive knowledge associated with the issue. It is associated with the fact that they did not include some important issues, while other were included even if they are not so crucial (e.g. sub-chapter “Modulation of GM by antidepressants”).

Author Response

We thank the reviewer for evaluating pur manuscript and in agreement with his suggestions we have:

1) added in the paragraph "Types, symptoms, and causes of depression" evidence about the involvement of GM dysbiosis in the onset of depression (please see lines105-110),

2) added in the paragraph "The role of the gut-brain axis in depression" evidence about the role of the GM in the modulation of  antidepressants responses (please see lines 131-135),

3) we have collapsed the paragraphs "Antidepressants Therapeutic Approaches" and "Gut microbiota and antidepressant therapies" (please see lines 265-286) in order to increase the manuscript fluidity.

However, as rightly observed by the reviewer we have not performed a systematic review but a literature review and the main reason is, considered the high novelty of this field of research, the absence in the currently literature of sufficient manuscripts for the writing a systematic review. Therefore, we cannot unfortunately include a Materials and Methods section because, as we did not write a systematic review, we did not follow a PRISMA flow diagram for study selection.

Round 2

Reviewer 1 Report

No other comments

Author Response

We thank the reviewer for the positive evaluation of our manuscript

Reviewer 2 Report

The manuscript entitled „ Bugs and drugs: how the intestinal microbiota modulates antidepressant therapies” presents interesting issue, but some problems should be corrected.

Authors prepared a literature review, which intended to answer a specific question – as defined within the title of the study “how the intestinal microbiota modulates anti-depressant therapies”, being a specific problem. However, within their manuscript, Authors presented a lot of information about either intestinal microbiota or depression/antidepressant therapies, but they did not answer this specific question.

As I indicated within my previous review, it may be observed even based on the sub-chapters, which are organized as follows:

(1)    Types, symptoms, and causes of depression (chapter about depression/antidepressant therapies, but not about intestinal microbiota)

(2)    The role of the gut-brain axis in depression (chapter about intestinal microbiota and depression/antidepressant therapies, but not answering the specific question about how the intestinal microbiota modulates anti-depressant therapies)

(3)    Antidepressants Therapeutic Approaches (chapter about depression/antidepressant therapies, but not about intestinal microbiota)

(4)    Gut microbiota and antidepressant therapies (chapter supposed to answer answering the specific question about how the intestinal microbiota modulates anti-depressant therapies, but as short as 15 lines and referring 6 references and not answering it)

(5)    Gut-microbiota based pharmacokinetics (chapter supposed to answer answering the specific question about how the intestinal microbiota modulates anti-depressant therapies, but as short as less than 1 page and not answering it)

(6)    Modulation of GM by antidepressants (the chapter about the reverse association – instead of how the intestinal microbiota modulates anti-depressant therapies describing how the anti-depressant therapies modulate intestinal microbiota)

After receiving my previous review, Authors in their response letter indicated that based on my comments they added 2 paragraphs:

(1)    “added in the paragraph "Types, symptoms, and causes of depression" evidence about the involvement of GM dysbiosis in the onset of depression (please see lines105-110)” – OK, but it still does not answer the question about how the intestinal microbiota modulates anti-depressant therapies

(2)    “added in the paragraph "The role of the gut-brain axis in depression" evidence about the role of the GM in the modulation of antidepressants responses (please see lines 131-135)” – great, but it is only 5 lines, so maybe Authors do not have adequate data to prepare such manuscript, if they are not able to answer the question about how the intestinal microbiota modulates anti-depressant therapies

The presented manuscript not only does not answer the specific question and the aim is not met, but also there are serious doubts associated with included references and general content.

The serious flaw of the presented manuscript is associated with the fact, that it presents a highly subjective review, not a systematic review. While the systematic review has a key role for broadening knowledge, the other reviews don’t have such role.

Taking into account, that the Materials and methods section is not presented (it should be added), without any specific information, it is hard to understand which studies were included into review and why. Authors did not present any key words, which were used during literature search, inclusion and exclusion criteria of references, information about the procedure of literature search conducted by them, number of chosen references, as well as information if some of them were excluded from the review and on the basis of which criteria. As a number of recent publications that are related to the issue were not included, it is a serious problem.

Authors do not present the current and comprehensive knowledge associated with the issue. It is associated with the fact that they did not include some important issues, while other were included even if they are not so crucial (e.g. sub-chapter “Modulation of GM by antidepressants”).

Taking this into account, it seems that Authors did not conduct the systematic review and as a result, they did not gather sufficient literature to answer the question about how the intestinal microbiota modulates anti-depressant therapies, and they present a lot of random information about depression and about microbiota, but do not answer this specific question.

Author Response

We thank the reviewer for the comments